# Risk of Social Isolation as a Contributing Factor to Diet Quality in Community-Dwelling Older Persons Living in the Australian Capital Territory—A Pilot Study

**DOI:** 10.3390/healthcare12050539

**Published:** 2024-02-24

**Authors:** Elizabeth Low, Nathan M. D’Cunha, Ekavi Georgousopoulou, Nenad Naumovski, Rachel Bacon, Stephen Isbel, Megan Brocklehurst, Matthew Reynolds, Daena Ryan, Jane Kellett

**Affiliations:** 1Discipline of Nutrition and Dietetics, Faculty of Health, University of Canberra, Canberra, ACT 2601, Australia; nathan.dcunha@canberra.edu.au (N.M.D.); ekavi.georgousopoulou@canberra.edu.au (E.G.); nenad.naumovski@canberra.edu.au (N.N.); rachel.bacon@canberra.edu.au (R.B.); jane.kellett@canberra.edu.au (J.K.); 2Centre for Ageing, Research and Translation, Faculty of Health, University of Canberra, Canberra, ACT 2601, Australia; stephen.isbel@canberra.edu.au; 3Functional Foods and Nutrition Research (FFNR) Laboratory, University of Canberra, Bruce, ACT 2617, Australia; 4University of Canberra Research Institute for Sport and Exercise (UCRISE), University of Canberra, Canberra, ACT 2601, Australia; 5Department of Nutrition and Dietetics, School of Health Science and Education, Harokopio University, Kallithea, 17671 Athens, Greece; 6Discipline of Occupational Therapy, Faculty of Health, University of Canberra, Canberra, ACT 2601, Australia

**Keywords:** older adults, social isolation, dietary inflammatory index

## Abstract

Objectives: Social isolation is recognised as a risk factor in the inflammatory process. This study explored the association between social isolation and the Dietary Inflammatory Index (DII) in community-dwelling older persons. Methods: This cross-sectional pilot study recruited 107 community-dwelling people aged over 55 years living in the Australian Capital Territory. Participants completed an extensive food frequency questionnaire and provided anthropometric and sociodemographic data. Social isolation was evaluated using the Lubben Social Network Scale (LSNS). Diet quality was assessed using DII. Results: Average age was 70.1 (±8.61) years and 62.8% were female. The average DII score was −1.10 (±1.21), indicating an anti-inflammatory diet. Higher LSNS was associated with lower DII (b (95% CI) = −0.041 (−0.066, −0.17); *p* < 0.01) and was positively influenced by the number of people in household (b (95% CI) = 5.731 (2.336, 9.127); *p* = 0.001). Conclusion: Increased risk of social isolation was associated with an increased tendency towards a more inflammatory diet. Reducing social isolation may decrease the inflammatory component of dietary intake for older persons living independently in the community.

## 1. Introduction

The role of nutrition as both a determinant of healthy ageing and a modifiable factor in maintaining a healthy phenotype is well established [1,2,3,4,5,6]. Equally well established is the role of non-dietary factors such as age-related physiological changes in the gut including early satiety [7,8], the diminution of senses of smell and taste [1,7,8], poor oral health [8,9], and several psychosocial factors, including social contact [7,8]. Additionally, the determinants of healthy eating for older persons are more than solely nutrition and involve complex interactions at both the individual and community levels [5,10,11,12] and a range of non-dietary factors [5,10,11,13,14,15] with a number of studies noting the association between increased risk of social isolation (and/or lack of social supports) with lower quality nutrition, including decreased intake of fruits and vegetables, reduced variety of foods, and lower energy intake [10,12,16,17,18,19,20].

Social isolation is increasingly becoming identified as a public health issue for older persons living in the community [13,21,22,23]. Factors contributing to social isolation include changed economic circumstances [22,24], declining physical health (including mobility challenges [22], loss of hearing and vision [13,25]), cognitive decline [26], death of a spouse [24], depression [13,22,27], and decreasing friendship circles [13,22,24]. Social isolation can also be associated with increased inflammatory response [28,29,30,31]. A study by Eisenberger (2016) summarises specific mechanisms demonstrating the role of social factors, such as social isolation, in immune system regulation including the impact of negative social factors that may result in pro-inflammatory responses [28,29,31]. There is a tendency for older persons to move physiologically to a pro-inflammatory state (inflammaging), [32,33,34,35,36] commonly associated with a number of ageing-related conditions such as dementia, sarcopenia, cardiovascular disease, Type 2 diabetes, and several cancers [36,37,38,39]. While not all studies are in agreement with the inflammatory markers involved, increased pro-inflammatory responses associated with negative social factors are reflected in increased levels of cytokines, C-reactive protein (CRP), and the upregulation of CD16 [28,29,30,31,40]. 

Compounding the negative effects of social isolation on inflammatory markers, the risk of social isolation has also been associated with poor nutritional health [16,17,19,41,42]. The impact on nutritional health due to social isolation includes increased risk of malnutrition [10,42], decreased fruit and vegetable intake [43], and decreased energy intake [10]. Poor dietary intake can result in inadequate nutrient intake and increased risk of malnutrition, with possible negative health consequences such as higher mortality risk [44,45], increased risk of sarcopenia [45] and falls [46], and poor wound healing [45,46]. Additionally, there is an established relationship between dietary intake and inflammatory response [40,47,48,49]. Diets high in whole grains, fruit and vegetables, nuts, and fish (such as the Mediterranean diet) have a positive association with lower concentrations of inflammatory markers such as CRP and tumour necrosis factor (TNF). This suggests a potential role of diet as a moderator of inflammatory response and as an important factor for healthy ageing [34,38,40,50]. The Dietary Inflammatory Index (DII) [49] is a validated dietary index assessing dietary intake with respect to the pro- and anti-inflammatory potential of food components. 

There is an increasing desire of older Australians to age in place, that is, independently in their own home and community [51,52]. Ageing in place, as a concept, has the capacity to provide care appropriate to the needs of older persons at a lower cost than residential facilities and enables older persons to maintain both quality of life and independence in familiar surroundings [53,54]. However, several studies identify several known risks associated with ageing in place, including the linked risks of malnutrition and social isolation [41,42,55]. Given the link between social isolation and inflammaging and social isolation and poor diet quality, using DII as a measure of diet quality, this pilot study aims to explore if an increased risk of social isolation is associated with the inflammatory nature of dietary intake for older community-dwelling persons. 

## 2. Methods

### 2.1. Sample

This was a cross-sectional pilot study of community-dwelling people aged 55 years and over living independently in the Australian Capital Territory (ACT). Participants were responders to advertisements on social media platforms, newsletters targeting older persons, and word of mouth (snowballing). Participants were eligible for inclusion if they were aged 55 years or older and living independently in the community in the ACT. No other exclusion factors were applied.

### 2.2. Ethics

The [University of Canberra] Human Ethics Committee approved this study prior to commencement (HREC #2079). Informed written consent was obtained from participants and all procedures were in accordance with the Declaration of Helsinki [56]. 

### 2.3. Participant Interviews

Participants attended a face-to-face interview (60–120 min) during which the following data were collected: a.Anthropometric and sociodemographic characteristics

Participants provided self-reported anthropometric data (height, weight, and waist circumference) and sociodemographic and lifestyle information (date of birth, education level, marital status, and number of people living in household). Body mass index (BMI) was calculated using self-reported data. 

b.Lubben Social Network Scale Score (LSNS)

The Lubben Social Network Scale Score (LSNS) is a validated survey tool specifically designed to assess the risk of social isolation in older populations [57,58]. This tool provides a composite measure of social connections amongst older persons with respect to contact with, and support provided by, family and friends. Interviewers administered the LSNS (12 questions) to measure social isolation risk. A minimum score of 0 (indicating high risk of social isolation) was possible, with maximum score of 60 (indicating low risk of social isolation). 

c.Dietary assessment

A validated and comprehensive 280-item Food Frequency Questionnaire (FFQ) [59,60,61,62] was completed where the frequency of consumption of a wide range of foods usually consumed over the previous 12 months was characterised. The FFQ additionally allowed free entry of foods/beverages not specifically covered. Data from the FFQ were analysed using Foodworks (v9; Xyris Software QLD, Brisbane, Australia).

### 2.4. Measure of Diet Quality—Dietary Inflammatory Index (DII) Score

From Foodworks data, the DII scores were calculated in Microsoft Excel 2022 (Microsoft Corporation, Redmond, WA, USA) using the prescribed food parameters and algorithm [49]. Using this algorithm, DII scores tend to lie within the range from −6 (anti-inflammatory) to +6 (pro-inflammatory) with a theoretical possible range being −9.0 to +8.0 [49]. This study used 39 of the possible 45 food parameters (Figure 1). The food parameters were determined through a literature-based review of foods and food constituents and how they affect specific inflammatory markers [48]. Calculation of DII requires energy intake in kilocalories; therefore, energy intake was converted from kilojoules to kilocalories.

### 2.5. Statistical Analysis

Statistical analyses were completed using IBM^®^ SPSS^®^ Statistics for Windows, Version 27 (IBM Corp., Armonk, NY, USA, 2021). Continuous variables are presented as mean and standard deviation and categorical variables are presented as frequencies and relative frequencies. All variables were evaluated to determine normality and suitability for parametric or non-parametric methods using histograms, Q-Q plots, and the Kolmogorov–Smirnov test of normality. Means were compared using the Student’s Test (or Mann–Whitney U-test) to determine if differences between groups were significant. Associations between categorical variables were explored using Pearson’s Chi-squared test (or Fisher’s exact test when appropriate). Pearson’s (or Spearman’s rho) correlation coefficient was used to explore linear relationships between continuous variables. Multivariable linear regression analysis was used to explore independent associations between variables, after adjusting for potential confounding factors. Results are presented as unstandardised beta coefficients, 95% confidence interval, and *p*-value (*p* < 0.05, *p* < 0.001). Linearity of models was tested through scatter plots of standardised residuals against standardised predicted values. Statistical significance was set at alpha = 5%.

## 3. Results

### 3.1. Participant Characteristics

Participants (n = 107) were 70.2 ± 8.6 years (*p* = 0.030) of age. Most participants (n = 73) were female (68.2%); 62 participants (57.9%) were married; 65 participants (60.07%) were living in a household with one or more other person/s. With respect to education levels, 104 participants (97.2%) had completed post-secondary education with 80 participants (74.8%) having completed tertiary education. Most participants were retired (n = 80, 74.8%), and 27 participants (25.2%) were working either on a full-time or part-time/casual basis. Participant characteristics are summarised in Table 1 and Table 2.

### 3.2. Dietary Intake

The average energy intake for all participants was 9297 ± 2249 kJ/day with intake ranging from 5254–16,091 kJ/day. Average energy intake varied between male and females and across household types. Average intake in one-person households was lower than in two-or-more-person households. However, the comparison of differences in means, based on sex and household type, found no statistical significance (*p* = 0.158 and 0.083, respectively). Average BMI was 25.19 ± 3.96 kg/m^2^ which is within an acceptable range for older persons. 

### 3.3. Diet Quality 

With respect to DII score, a more negative score indicates a less inflammatory diet. The mean DII score for all participants (−1.10 ± 1.2, *p* = 0.018) showed a tendency towards an anti-inflammatory diet; however, the overall range of scores is wide (−3.46 to +3.66, *p* = 0.030) with the widest range (−3.46 to +3.66) occurring in single-person households (Table 2). The comparison of DII means by household type and by sex found that these were associated with a less inflammatory score (*p* = 0.030 and *p* = 0.018, respectively). Univariable regression modelling found that LSNS negatively influenced DII score for all participants (b (95% CI) = −0.041 (−0.066, −0.17); *p* < 0.01) and explained 9.6% score variation amongst participants. Multiple regression modelling (Model 3, Table 3) adjusting for household type, sex, and age found that for every one unit increase in LSNS, DII will decrease (become less inflammatory) (b (95% CI) = −0.032 (−0.057, −0.006), *p* < 0.001), explaining 17.7% of score variation amongst participants. Further, Model 3 suggests that being female also influenced predicted DII score (b (95% CI) = −0.545 (−1.037, −0.053); *p* < 0.001) (Models 5 and 6). The univariate modelling of sex and DII shows that (b (95% CI) = −0.588 (−1.073, −0.102); *p* = 0.018), and R^2^ = 0.052.

### 3.4. Lubben Social Network Scale Scores

The average participant LSNS score was 38.7 ± 9.06 with a range of 7–58 (Table 2) suggesting that participants were generally not at high risk of social isolation. The average LSNS score for participants living in two-or-more-person households was higher than for participants in a single-person household (*p* = 0.001). No statistical significance for differences in means based on sex (*p* = 0.434) was observed. Multiple regression modelling adjusting for household type, sex, age, and BMI was undertaken (Table 4). Only household type was associated with an increased LSNS score (*p* < 0.001). 

## 4. Discussion

There are a number of studies that support the association between inflammation, the ageing process, and the quality of ageing (inflammaging) and supporting the association between social isolation and the presence of inflammation. Additionally, older persons, particularly those ageing in the community, are at increased risk of social isolation, which is associated with both an increased risk of inflammation and reduced nutrition. The aim of this pilot project was to evaluate whether the risk of social isolation influenced the inflammatory quality of dietary intake for older independent community-living persons. The results suggest that the predicted DII score will be reduced by −0.041 units (indicative of a less inflammatory diet) for every unit increase in LSNS score (higher LSNS scores indicating lower risk of social isolation). The interactions between the risk of social isolation, inflammaging, and diet quality are multi-faceted and complex (refs [16,34,42]). However, the results of this pilot study suggest that lower risk of social isolation (as measured by LSNS) may be a contributory factor for persons following a less inflammatory diet. 

While not specific to DII, these results are supported by several studies showing the negative impact of social isolation on dietary intake in older independent community living persons [10,11,15,63], predominantly through the impact on fruit and vegetable intake [10,64]. Additionally, studies have reported an association between dietary intake, sex, and social isolation, finding that men who are living alone, poorly supported socially, or experiencing social isolation are more nutritionally vulnerable [9,10,15] and consuming fewer servings of fruits and vegetables [10,15]. 

This pilot study additionally found that sex influenced the predicted DII score (−0.228, (CI = −1.073, −0.102), *p* = 0.018) for women. However, the influence was small, explaining only 5.2% of the sample variation. One possible explanation for the small effect relates to the self-selection of participants for the study, potentially reflecting an existing interest in nutrition coupled with the high level of education for most participants, suggesting that male participants may have a level of health literacy that may not be reflected in a larger population of older men [14,65]. Further, most male participants (73.5%) were married, with several studies noting the positive effects on nutritional intake experienced by married/partnered men [10,11,12,66]. 

One of the limitations of this study is that only risk of social isolation was measured. Several studies suggest that social isolation is more than a lack of social networks and poor social participation but includes an individual’s perception of isolation (expressed through feelings of loneliness) [27,28,67,68]. For example, an individual may have few social interactions and a limited social network but may not consider themselves lonely or experience feelings of social isolation. It is possible that some participants with low LSNS scores may not perceive themselves as at risk of social isolation, and some participants with higher LSNS scores may perceive the opposite. A study undertaken by Boulos [42] identified loneliness and social isolation as independent risk factors with respect to malnutrition among older persons. Future studies need to consider participants’ perception of social isolation to determine if the perception of social isolation, rather than risk, impacts DII scores, for example, by using the DeJong Giervald Loneliness Scale [69,70]. 

Additional limitations of this pilot include sex balance, self-reported data, and the single focus on food components relating to calculation of DII scores. Given that sex is known to influence nutritional behaviours and status amongst older persons, the greater number of female participants may have positively influenced DII results [9,10,15]. Further, most male participants were married, with marital status having a beneficial impact on nutritional status for males [9,10,15]. The self-reporting of anthropometric and food intake data for dietary assessment is a known limitation [71,72,73]. Weight and BMI are often under-reported [71], as are energy and fat intake [72,73]. To some extent, this limitation was mitigated through the design of the food frequency questionnaire to accommodate the free entry of food items not listed [73]. However, this remains a limitation of the study. While this pilot study focused on exploring the association between diet quality (as measured by DII) and the risk of social isolation, future studies should consider the impact of other modifiable lifestyle factors such as physical activity, smoking, and supplement intake. 

The strengths of this study include its novelty in this population, its use of a validated measure of diet quality, and its contribution to the limited pool of nutrition studies investigating the link between diet quality and healthy ageing with respect to older persons ageing in place. Further analysis should also be undertaken to determine if specific dietary patterns/food choices were more prominent in those with a higher risk of social isolation, contributing to a more inflammatory nature of dietary intake. 

## 5. Conclusions 

The results of this pilot study suggest that a decreased risk of social isolation is associated with a less inflammatory diet in community-dwelling people aged 55 years or over, as measured by DII, with female participants having a lower DII score (less inflammatory). While the effects are small, our findings provide some evidence identifying the association between diet quality and the risk of social isolation in this population. These findings support that diet should be considered as an important pillar for healthy ageing and promoting the role of anti-inflammatory diets is an important component to modulate inflammaging in older persons who may be at risk of social isolation.

## Figures and Tables

**Figure 1 healthcare-12-00539-f001:**
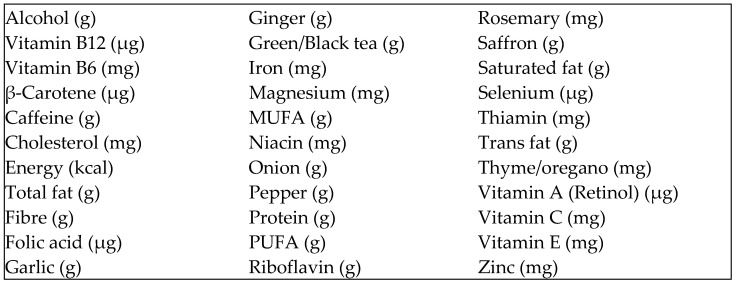
Food parameters used to calculate DII score.

**Table 1 healthcare-12-00539-t001:** Participant statistics summary by sex (n = 107).

	All	Male (M)	Female (F)	*p*-Value
Number of Participants (n)	107	34 (31.8%)	73 (68.2%)	
Age (years) ^1^:				
Mean	70.2 ± 8.6	72.8 ± 8.2	68.9 ± 8.5	
Range	56–94	60–94	56–88	0.030
Marital Status:				0.067
Married	62 (57.9%)	25 (73.5%)	37 (50.7%)	
Not Married—All	45 (42.1%)	9 (25.5%)	36 (49.3%)	
Never Married	9	1	8	
Divorced	15	1	14	
Widowed	16	6	10	
Other	5	1	4	
Employment Status				0.494
Retired	80 (74.8%)	25 (73.5%)	55 (75.3%)	
Employed—All	27 (25.2%)	9 (26.5%)	18 (24.7%)	
Full-time	12	3	9	
Casual	9	4	5	
Part-time	5	1	4	
Unemployed	1	1	0	
Education				0.102
Year 10 only	3 (2.8%)	3 (8.8%)	0	
Post-secondary—All	104 (97.2%)	31 (91.2%)	73 (100%)	
Year 12	7	1	6	
Certificate/Diploma	17	6	11	
Undergraduate	52	15	37	
Post-graduate	28	9	19	
BMI (kg/m^2^) ^1^	25.19 ± 3.96	26.03 ± 3.73	24.79 ± 4.03	
Range	16.87–39.66	20.90–38.01	16.87–39.66	0.133
Energy—Daily (kJ) ^1^	9297 ± 2249	9748 ± 2259	9087 ± 2228	0.158
Range	5254–16,091	5327–14,643	5254–16,091	
DII Score ^1^	−1.10 ± 1.21	−0.70 ± 1.43	−1.28 ± 1.04	
Range	−3.46–+3.66	−2.70–+3.66	−3.46–0.85	0.018
LSNS Score ^1^	38.7 ± 9.06	37.8 ± 7.58	39.0 ± 9.68	
Range	7–58	20–53	7–58	0.434

^1^ Values shown as mean ± SD. Note: DII—Dietary Inflammatory Index; LSNS—Lubben Social Network Scale.

**Table 2 healthcare-12-00539-t002:** Participant statistics summary by household type.

	All	Single Person Household	2 or More Person Household	*p*-Value
Number of Participants (n=)	107	42 (39.3%)	65 (60.07%)	
Age (years) ^1^:				
Mean	70.2 ± 8.6	73.2 ± 8.9	68.2 ± 7.8	
Range	56–94	58–94	56-84	0.003
Sex:				0.065
Male	34 (32%)	9 (26%)	25 (74%)	
Female	73 (68%)	33 (45%)	40 (55%)	
Marital Status:				<0.001
Married (incl de facto)	62 (57.9%)	3 (7.2%)	59 (90.8%)	
Not Married	45 (42.1%)	39 (92.8%)	6 (9.2%)	
Never Married	9	9	0	
Divorced	15	12	3	
Widowed	16	15	1	
Other	5	3	2	
Employment Status:				0.050
Retired	80 (74.8%)	36 (85.7%)	44 (67.1%)	
Employed	27 (25.2%)	6 (14.3%)	21 (32.3%)	
Full-time	12	3	9	
Casual	5	3	2	
Part-time	9	0	9	
Unemployed	1	0	1	
Education:				0.656
Year 10 only	3 (3%)	1 (2.4%)	2 (3.1%)	
Post-secondary	104 (97%)	41 (97.6%)	63 (96.9%)	
Year 12	7	4	3	
Certificate/Diploma	17	5	12	
Undergraduate	52	19	33	
Post-graduate	28	13	15	
BMI (kg/m^2^) ^1^	25.19 ± 3.96	25.26 ± 4.19	25.14 ± 3.84	
Range	16.87–39.66	16.87–39.66	18.27–38.01	0.879
Energy—daily (kJ) ^1^	9297 ± 2249	8827 ± 2441	9600 ± 2078	
Range	5254–16,091	5254–16,091	5327–14,065	0.083
DII Score ^1^	−1.10 ± 1.21	−0.79 ± 1.20	−1.30 ±1.17	
Range	−3.46–+3.66	−3.46–+3.66	−3.23–+2.38	0.030
LSNS Score ^1^	38.7 ±9.06	35.2 ± 9.18	40.9 ± 8.23	
Range	7–58	7–48	20–58	0.001

^1^ Values shown as mean ± SD. Note: DII—Dietary Inflammatory Index; LSNS—Lubben Social Network Scale.

**Table 3 healthcare-12-00539-t003:** Regression results for Lubben Social Network Scale Score: unstandardised B (95% Confidence Interval).

Parameter	Unadjusted		Model 1		Model 2		Model 3		Model 4	
	b	95% CI	b	95% CI	b	95% CI	b	95% CI	b	95% CI
Number of people in household (2 or more persons vs. 1 person)	5.731 *	2.336–9.127	6.159 *	2.722–9.595	6.956 **	3.320–10.592	7.204 **	3.583–10.825	7.200 **	3.558–10.842
Sex (Female vs. Male)			2.510	−1.094–6.114	3.188	−0.552–6.927	3.177	−0.533–6.888	3.161	−0.609–6.931
Age (per 1 year increase)					0.137	0.072–0.346	0.172	−0.040–0.384	0.172	−0.041–0.3
Education level (Tertiary vs. Non-tertiary)							3.155	−0.717–7.026	3.155	0.381–20.792
BMI (per 1 kg/m^2^ increase)									−0.012	−0.434–0.400

* Significant at *p* < 0.05 level. ** Significant at *p* < 0.001 level. Note: BMI—Body Mass Index.

**Table 4 healthcare-12-00539-t004:** Regression Results for Dietary Inflammatory Index: Unstandardised B (95% Confidence Interval).

Parameter	Unadjusted		Model 1		Model 2		Model 3		Model 4		Model 5	
	b	95% CI	b	95% CI	b	95% CI	b	95% CI	b	95% CI	b	95% CI
LSNS Score (per 1 unit increase)	0.041 *	−0.066–−0.017	−0.036 **	−0.062–−0.010	−0.032 *	−0.057–−0.006	−0.033 *	−0.059–−0.008	−0.030 *	−0.055–−0.004	−0.030 *	−0.055–−0.004
No. People in household (2 or more persons vs. 1 person)			−0.309	−0.782–0.165	−0.441	−0.913–0.030	−0.338	−0.842–0.166	−0.397	−0.901–0.106	−0.395	−0.901–0.112
Sex (Female vs. Male)					−0.627 *	−01.099–−0.156	−0.545 *	−1.037–−0.053	−0.555 *	−1.042–−0.067	−0.543	−1.038–−0.048
Age (per 1 year increase)							0.016	−0.012–0.43	0.010	−0.017–0.038	0.010	−0.018–0.038
Education (Tertiary vs. Non-tertiary)									−0.440	−0.945–0.065	−0.444	−0.952–0.064
BMI (per 1 kg/m^2^ increase)											0.009	−0.046–0.064

* Significant at <0.05 level. ** Significant at *p* < 0.001 level. Note: LSNS—Lubben Social Network Scale; BMI—Body Mass Index.

## Data Availability

Data are contained within the article.

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
