# Peer review of "Risk of Social Isolation as a Contributing Factor to Diet Quality in Community-Dwelling Older Persons Living in the Australian Capital Territory—A Pilot Study"

_healthcare, 2024, doi:10.3390/healthcare12050539_

Round 1

Reviewer 1 Report

Comments and Suggestions for Authors

The article presented is very well written, clear and understandable. The research design is comprehensive established. The methodology is relevant and appropriate. The results are stated well and are a good contribution to the scientific community. The discussion is correct and appropriate use has been made of bibliographical references. However, I consider that there are some details that need to be improved.

I think that the relationship between social isolation and increased inflammatory response needs to be better explained. In the second paragraph of the introduction biomedical aspects are mentioned, but it is not specified whether social isolation is related to the pro-inflammatory state due to a reduced mobility, to a favourable socio-psychological phenomenon or to any other social phenomena. It is important to clarify this aspect so that, in the event that the article could be read by policy makers, the strategies and social measures to be taken could be improved. Similarly, in the third paragraph of the introduction it is mentioned that social isolation is related to poor nutritional health. Why? It is important that the authors do not take knowledge for granted and clarify the claims.

I would recommend modifying the tables by using smaller font and eliminating line spacing.

Line 208 states: "That is, people with a lower risk of social isolation (as measured by LSNS) may have a less inflammatory diet".

I would like to humbly comment to the authors on an epistemological error (determinism) they make in this statement. Although the statement is posed as a possibility, it is asserting that the absence of social isolation by itself may imply a less inflammatory diet. This is completely incorrect. Social isolation has implications on people's lives: mobility, relationships, perceptions, etc. Likewise, social isolation is also related to a certain construction of society. All this means that social emotions are constructed in such a way that people can stop engaging in healthy behaviours and, of course, vice versa. Hence, the statistical relationship can be better explained. However, as I say, the above statement does not fit well with current knowledge. So I suggest you change it and avoid determinism as much as possible. I am aware that they clarify this aspect adequately later in the limitations of the study. However, I still consider these statements to be inadequate in the light of current knowledge (to which the authors precisely refer).

Author Response

The article presented is very well written, clear and understandable. The research design is comprehensive established. The methodology is relevant and appropriate. The results are stated well and are a good contribution to the scientific community. The discussion is correct and appropriate use has been made of bibliographical references. However, I consider that there are some details that need to be improved.

Thank you for your kinds work and the time taken.

I think that the relationship between social isolation and increased inflammatory response needs to be better explained.

In the second paragraph of the introduction biomedical aspects are mentioned, but it is not specified whether social isolation is related to the pro-inflammatory state due to a reduced mobility, to a favourable socio-psychological phenomenon or to any other social phenomena. It is important to clarify this aspect so that, in the event that the article could be read by policy makers, the strategies and social measures to be taken could be improved.

Similarly, in the third paragraph of the introduction it is mentioned that social isolation is related to poor nutritional health. Why? It is important that the authors do not take knowledge for granted and clarify the claims.

This paragraph has been amended in line with suggestions.

This paragraph has also been amended in line with suggestions

I would recommend modifying the tables by using smaller font and eliminating line spacing.

Tables have been amended

Line 208 states: "That is, people with a lower risk of social isolation (as measured by LSNS) may have a less inflammatory diet".

I would like to humbly comment to the authors on an epistemological error (determinism) they make in this statement. Although the statement is posed as a possibility, it is asserting that the absence of social isolation by itself may imply a less inflammatory diet. This is completely incorrect. Social isolation has implications on people's lives: mobility, relationships, perceptions, etc. Likewise, social isolation is also related to a certain construction of society. All this means that social emotions are constructed in such a way that people can stop engaging in healthy behaviours and, of course, vice versa. Hence, the statistical relationship can be better explained. However, as I say, the above statement does not fit well with current knowledge. So I suggest you change it and avoid determinism as much as possible. I am aware that they clarify this aspect adequately later in the limitations of the study. However, I still consider these statements to be inadequate in the light of current knowledge (to which the authors precisely refer).

We appreciate your drawing this to our attention. This sentence has been modified . 

Reviewer 2 Report

Comments and Suggestions for Authors

Dear Authors,

Please find some suggestions to your manus below:

Title:

considering that this is a cross-sectional study, without the possibility to infer causality, authors should avoid the expression influencing ("Risk of social isolation as a factor influencing diet quality in community-dwelling older persons living in the Australian Capital Territory – a pilot study), using for example, correlates, association, etc.)

Abstract:

In the section of the abstract, could the authors increase clarity (especially for those readers not familiar with the score of LSNS) in terms of "higher/lower social isolation" instead of just mentioning " LSNS was associated...":

"Social isolation was evaluated using the Lubben Social Network Scale 35

(LSNS). Diet quality was assessed using DII. Results: Average age was 70.1 (±8.61) years and 62.8% 

were female. The average DII score was -1.10 (±1.21), indicating an anti-inflammatory diet. LSNS

was associated with DII (b (95% CI): -.041 (-.066, -.17); p<.01), and was positively influenced by

number of people in household (b (95% CI) =5.731, (2.336, 9.127; p=0.001)."

Methods:

Did the authors have collected information on physical activity, nutritional supplements (and medication), and tobacco use to address the association between social isolation and diet quality? If not, this should be considered and acknowledged as a strong limitation of the study.

Do the authors have any data on the validity of self-reported anthropometric data for this population? If not, this should be taken specifically into account in the limitations.

Considering that diet quality is a broad expression encompassing many different dimensions, and authors looked only at DII, this should be defined ("diet quality") in the methods, and if only DII was measured, then authors should use DII instead of diet quality.

Clarify gender and sex along the text.

Results:

Authors should provide descriptive data for food and nutritional parameters used to calculate the DII score (combining sections 3.2 and 3.4).

Discussion:

Revise according to the above.

Conclusion:

lines 251-254 could be moved to the discussion section.

Author Response

Dear Reviewers

Thank you for taking the time to review our paper and for the valuable feedback.

Please find our response to you very constructive comments.

Our response to your suggestions is noted below. 

Reviewer 1:

Title:

considering that this is a cross-sectional study, without the possibility to infer causality, authors should avoid the expression influencing ("Risk of social isolation as a factor influencing diet quality in community-dwelling older persons living in the Australian Capital Territory – a pilot study), using for example, correlates, association, etc.)

Title has been amended in line with your suggestion.

Abstract:

In the section of the abstract, could the authors increase clarity (especially for those readers not familiar with the score of LSNS) in terms of "higher/lower social isolation" instead of just mentioning " LSNS was associated...":

"Social isolation was evaluated using the Lubben Social Network Scale 35 (LSNS). Diet quality was assessed using DII. Results: Average age was 70.1 (±8.61) years and 62.8% were female. The average DII score was -1.10 (±1.21), indicating an anti-inflammatory diet. LSNS was associated with DII (b (95% CI): -.041 (-.066, -.17); p<.01), and was positively influenced by number of people in household (b (95% CI) =5.731, (2.336, 9.127; p=0.001)."

Amended as suggested.

Amended as suggested.

Methods:

Did the authors have collected information on physical activity, nutritional supplements (and medication), and tobacco use to address the association between social isolation and diet quality? If not, this should be considered and acknowledged as a strong limitation of the study.

Do the authors have any data on the validity of self-reported anthropometric data for this population? If not, this should be taken specifically into account in the limitations.

Considering that diet quality is a broad expression encompassing many different dimensions, and authors looked only at DII, this should be defined ("diet quality") in the methods, and if only DII was measured, then authors should use DII instead of diet quality.

Clarify gender and sex along the text.

As the focus was on diet, this data was not included.  Considered and noted as a limitation.

Noted as a limitation.

DII has been more clearly defined as the measure of diet quality (line 96).

Thank you for the comment. This has been adjusted as per suggestion.

Results:

Authors should provide descriptive data for food and nutritional parameters used to calculate the DII score (combining sections 3.2 and 3.4).

Additional description of the food parameters used in the calculation of the DII has been included.  Because the focus of the pilot was to determine if there was an association, no additional analysis was undertaken on food parameters.  This will be considered for future studies.

Discussion:

Revise according to the above.

Amended as suggested.

Conclusion:

lines 251-254 could be moved to the discussion section.

Lines have been moved.

Round 2

Reviewer 2 Report

Comments and Suggestions for Authors

Although some limitations in the study exist, as stated by the authors, the questions raised in the present review were adequately addressed.